# Correlation Between Blood Coagulation Profile and Viscosity: Clinical Laboratory Observational Study

**DOI:** 10.3390/medsci13010020

**Published:** 2025-02-16

**Authors:** Ezekiel U. Nwose, Phillip T. Bwititi

**Affiliations:** 1School of Health & Medical Sciences, University of Southern Queensland, Toowoomba, QLD 4350, Australia; 2School of Dentistry & Medical Sciences, Charles Sturt University, Wagga Wagga, NSW 2650, Australia; pbwititi@csu.edu.au

**Keywords:** blood viscosity, D-dimer, eWBV, haemostasis indices, platelet count

## Abstract

Whole blood viscosity is a test for blood stasis and is an ideal evidence-based pathology parameter that is largely undervalued and retrogressing in clinical utilization. Coagulation profiles as indices of haemostasis are available but limited to central or referral laboratories and often involve long turn-around time. It is therefore important to study the correlation between the index of stasis and indices of haemostasis. Objective: To investigate the correlation of index of stasis with indices of haemostasis. Method: The clinical laboratory observational research method, using archived pathology data. Indices of haemostasis including activated partial thromboplastin time (APTT) and prothrombin time (PT), the international normalization ratio (INR), and plasma D-dimer were evaluated. On the other hand, the index of blood stasis used was the estimated whole blood viscosity (eWBV) and derived haematocrit and serum protein levels. All (N = 193) tests were collected within a calendar year from the same pathology service, and further, for the correlation, each set of variables from the same blood sample collection was used. Results: The haemostasis data are skewed (skewness > 2.0), while eWBV and platelet count are normal (skewness < 2.0). Haemostasis indices have an inverse association with eWBV (*p* < 0.001). The concordance and correlation of eWBV with platelet count is positive, weak, and significant (*p* < 0.001), but negative and negligible with PT and APTT. Conclusion: There are limitations to the possible correlation between eWBV and haemostasis indices. However, haemostasis indices have inverse associations with eWBV, and the latter can aid in the evaluation of haemostasis hence could be utilized as an alternative or complementary test to haemostasis tests. Research may normalize skewed data to obtain better correlation; therefore, further study is required to advance discourse, giving cognizance to clinical practice.

## 1. Introduction

Haemostasis has been an established medical concept, and the pathophysiology continues to constitute a clinical management challenge [1,2,3]. In the last 30 years, medical science research has demonstrated a possible relationship between haemostasis and dyslipidaemia [4], antiplatelet medication [5,6,7,8], and the trios of vasculopathy known as Virchow’s triad [9]. However, there is still a need for the accurate and validated prediction of bleeding risk [10,11].

It suffices to say haemostasis is a physiological balance between fibrinolysis and coagulation, or a central point of preventing bleeding and thrombosis. Hence, pathology tests in the clinical management of patients with haemostasis comprise mainly full blood count and coagulation profiles, especially including but not limited to platelet count, prothrombin time, and activated partial thromboplastin time. In more resourced settings and research centres, the plasma D-dimer test is carried out to assess the magnitude of fibrinolysis that in turn reflects the extent of coagulation [12]. It is important to emphasize that the plasma D-dimer test has been of interest but limited by problems of affordances [13,14,15].

In defining haemostasis as a balance between coagulation and fibrinolysis, there is an associated pathophysiology on either side of the continuum including bleeding and thrombosis. In the concept of Virchow’s triad, the pathophysiology is underpinned by endothelial dysfunction, hypercoagulation, and blood pooling or stasis [9], whereby the clinical pathology parameters are homocysteine, plasma D-dimer, and blood viscosity, respectively [16]. Although, endothelial dysfunction impacts blood stasis, hence, homocysteine is sometimes measured as a factor that affects blood viscosity [17]. None of these pathology tests are accessible in rural and remote settings, especially in low–mid-income countries where healthcare resources are limited.

Estimated whole blood viscosity (eWBV) is one test that has been advanced to extrapolate the value from routine haematocrit and serum protein [18]. Indeed, the eWBV dates back over 30 years [19] and has undergone validation processes including previous studies that have demonstrated the applicability of the medical science theory in diabetes [20], as well as proof of concept [16,17,21,22]. Nevertheless, the blood viscosity test is still overlooked in clinical practice [23,24,25], perhaps due to limited interpretation that tends to focus on hyperviscosity syndrome [26,27,28,29]. It is interesting to note that even where the test has been performed, albeit in a reference laboratory [30], the service has seized because it is erroneously viewed to be of little or no value [29]. Thus, the implication is that an established medical science has retrogressed.

Therefore, there is a need to advance the knowledge with a view to reintroduce blood viscosity testing into clinical practice. Motivated by recent reports on gender differences in coagulation parameters [31], and our previous reports [16,21,22,32]; the objective of this study is to briefly evaluate the correlation between the laboratory parameters of haemostasis and blood stasis (eWBV).

## 2. Materials and Methods

**Design:** This was a clinical laboratory observational study. Evaluation was designed to include descriptive approach, as well as comparisons and correlation. The correlation evaluation was performed to observe epidemiological factors [33]. The study design can also be described as a cross-sectional observation or clinical audit, with a small cohort evaluation using archived clinical pathology data.

**Setting:** Pathology data from regional New South Wales of Australia, as previously published [16,21,22].

**Data variables:** Parameters of haemostasis were limited to prothrombin time (PT) and activated partial thromboplastin time (APTT) as well as plasma D-dimer as in the study of He et al. [31].

**Inclusion criteria**: The selection criteria were limited to cases with a complete set of results for haematocrit, serum protein, and plasma D-dimer, as well as APTT, PT INR, and platelet count.

**Method of eWBV measurement:** The blood viscosity in this study was determined by the extrapolation method, as previously published [34]. It is pertinent to note that this eWBV option was compared with the digital method used in a reference laboratory [30].

**Statistical analysis:** Data were analyzed to check for differences in age and gender groups in consideration of other studies [26]. Considering the likelihood of fibrinolysis impacting pre-menopausal women [35] and previous observations in our study [16], the D-dimer variable was used to categorize data into [yes] and [no] dichotomous groups to compare the changes in eWBV with haemostasis indices (APTT, PT, and INR) and platelet count. Lastly, the levels of correlations among the eWBV, APTT, PT INR, and platelet count were evaluated using the Pearson correlation method.

**Limitations:** Cognizance was given to the sample size being considered relatively small. Therefore, “practical strategies for getting the most out of data when sample size is small” were adopted [36]. This included the measurement of theoretically strong indicators (APTT, D-dimer, and PT), as well as including a covariate (INR, another independent variable) that has been previously determined to be strongly related to the eWBV.

## 3. Results

For this cross-sectional study, a sample size of N = 193 satisfied the inclusion criteria. Descriptive statistics show that haemostasis data are asymmetrically distributed (kurtosis > 3.0 and skewness > 2.0), while the eWBV is symmetrical (Table 1).

***Comparison of the variables between stratified age groups***: Haemostasis variables and platelet count decreased at the 31–45 years of age group compared to the ≤30 years of age group, but increased thereafter, and only APTT achieved statistical significance (*p* < 0.04). The opposite is the case with serum proteins and eWBV in terms of directional change, but with a greater level of significance (Table 2).

***Comparison of gender groups***: The haemostasis variables are higher in women but not statistically significant, except for platelet count (*p* < 0.02). The opposite is the case with serum proteins and eWBV in terms of directional change and with a greater level of statistical significance (Table 3).

***Comparison of eWBV and haemostasis variables between dichotomous D-dimer groups***: APTT is statistically significantly higher among those with the positive group (*p* < 0.03). Conversely, eWBV and its indices are significantly lower (*p* < 0.001) compared to the negative group. The average age is noted to be higher with the positive D-dimer group (*p* < 0.001, Table 4).

***Correlation and cohort analyses***: It is observed that eWBV is moderately positively correlated with platelet count (r −0.26, *p* < 0.001), but negligibly negative with the haemostasis indices (r < 0.20, Table 5). Among the data, 16/193 comprised a cohort of n = 8 record numbers with repeated tests (Table 6). A concordance review of the small cohort shows a 100% association between eWBV and platelet count. The concordance of eWBV was in 3/8 and varied in the other 5/8.

## 4. Discussion

Table 1 shows that haemostasis data are quite asymmetrically distributed (kurtosis > 3.0 and skewness > 2.0), while eWBV and platelet counts are symmetrical. The evaluation of correlation for normally distributed data such as eWBV and platelet counts would require Pearson’s method. The asymmetric haemostasis indices would require log transformation, which can be conducted in research but not in a routine clinical pathology practice.

It is common in clinical practice that some data are asymmetric [37], and given that data are interpreted as they are in clinical practice, correlations between eWBV and haemostasis indices may be spurious. In the stratified age groups, haemostasis variables decreased while eWBV increased, and the latter showed a greater level of statistical significance (Table 2). There is indication of inverse association subject to verification with correlation analysis.

A previous study has reported the inverse relation of eWBV with INR [32]. What this report contributes is that similar inverse relationships may be occurring between eWBV and haemostasis indices (PT and APTT). The algorithmic eWBV test may be an alternative or complementary test to PT and APTT, especially where the turn-around time of the haemostasis test is a consideration.

The levels of haemostasis indices are insignificantly lower in men, while blood viscosity is significantly higher in men, compared to women (Table 3). The inverse relationship between eWBV and haemostasis indices is noteworthy given that fibrinolysis is lower among men than albeit pre-menopausal women [35]. Haemostasis is physiological balanced between coagulation and fibrinolysis, and it suffices that the lower rate of fibrinolysis explains the lower haemostasis biomarkers in men.

This contributes empirical data to support the higher levels of PT and APTT in women than men. Considering menstrual cycles and lower haematocrit in women, it is expected that eWBV would be higher in men [38]. Hence, the tendency for a higher risk of venous thrombosis among men than women [39,40]. On the other hand, there is reported potential of blood stasis being higher among women [41]. This may be a confound risk of the higher fibrinolysis being associated with greater plasma D-dimer in women and lower among men [42].

In the dichotomous D-dimer groups, APTT is significantly higher among those with the positive group (*p* < 0.03), while age as well as eWBV and its indices are significantly lower (*p* < 0.001), relative to the group that tested negative for D-dimer. Platelet count is also lower in the positive D-dimer group, although statistical significance is not achieved in this dataset (Table 4). This observation further reaffirms preceding observations, indicating inverse relations.

There is evidence that positive D-dimer is associated with anemia [43], and further, enhanced haemostasis activities indicated by increased plasma D-dimer are associated with thrombocytopenia [44]. Therefore, there is no gainsaying that blood viscosity could be inversely related with the haemostasis physiological process. Further, higher APTT is associated with positive D-dimer in this report, and this observation agrees with the report of another observational study that plasma D-dimer may increase with APTT and PT in certain circumstances [45]. Yet, APTT is expected to decrease, while D-dimer increases [46]. There is the idea that APTT can complement D-dimer in clinical management [46,47], hence, this report advances that eWBV is a valuable pathology biomarker to consider.

The correlation analysis confirms the preceding observations, i.e., that blood viscosity is negatively associated with haemostasis indices but positively associated with platelet count (Table 5). This is further confirmed by concordance review, as eWBV consistently increased with platelet count and vice versa in the subset of data (Table 6). Perhaps, it can be inferred and incontrovertible to note that while ‘stasis’ is a suffix in haemostasis, and the two terms are related, there is little difference in physiological phenomena, hence, laboratory tests that specifically evaluate for the two phenomena. It is also noteworthy that eWBV and platelet count are symmetrical data, while the haemostasis profiles are skewed; therefore, there is a strong limitation and reason for negligible correlation.

Antiplatelet therapy and anticoagulant medications are blood thinners in cardiology management, but the mechanisms of action are different. Anticoagulant is focused on the coagulation cascade and blood clot such as warfarin actin on the vitamin K pathway, while the antiplatelet therapy works on platelet function and stasis [48]. Blood viscosity is more specifically related to blood pooling; hence, it correlates with blood flow [49].

### Significance in Clinical Practice

There are empirical data showing that INR and WBV are inversely related [32,49]. This report contributes additional empirical data on other test parameters of haemostasis. To our knowledge, this is the first observational study to focus on the correlation between blood viscosity and the clinical pathology-based tests of haemostasis. More importantly, this report highlights the limitation of the correlation. There may be a temptation in research to log-normalize the skewed data, but clinical practice should be the guide.

It is also pertinent to highlight that the issue of bleeding remains a concern [50,51,52]. The therapeutic management still constitutes a dilemma [53,54,55,56], hence the continued research interest [57]. The significance for clinical practice cuts across several health issues involving cardiovascular complications, including but not limited to diabetes and kidney disease [12,57]. It has long been known that blood viscosity can be used to assess polycythemia complications [58], including but not limited to retinopathy [59].

Therefore, beyond the contribution of empirical data to advance the medical science of blood stasis, this report advances eWBV as an alternative or complimentary pathology test applicable to a wide range of disease management. For instance, the thromboembolic risk score assessment model is still questionable, hence limited in usage [60]. Further studies would be necessary to compare eWBV as a hemorheology predictor versus the risk score model in the assessment of future thromboembolic events.

## 5. Conclusions

This study identified haemostasis management as one area of clinical practice where laboratory methods would benefit from eWBV as a pathology test. The correlations between eWBV and the haemostasis indices (APTT, INR, and/or PT) constitute an advancement in knowledge regarding the science for improved healthcare practice. The method can be adopted and replicated. Thus, the relevance lies in the re-evaluation of eWBV for adoption and utilization as either an alternative or complementary test to the profile of APTT, INR, and/or PT.

## Figures and Tables

**Table 1 medsci-13-00020-t001:** Descriptive statistics of dataset.

	PROT	HCT	eWBV	INR	PT	APTT	PLT	Age
Mean	68.24	0.38	15.86	1.38	15.85	30.65	223.8	57.62
Standard Error	0.87	0.00	0.17	0.05	0.56	0.73	7.13	1.52
Median	69	0.389	16.17	1.1	13	27	234	61
Mode	72	0.417	16.50	1.1	12	27	292	78
Standard Deviation	12.08	0.06	2.39	0.68	7.75	10.13	98.77	21.12
Sample Variance	145.86	0.00	5.70	0.46	60.06	102.57	9754	446.10
Kurtosis	3.07	0.08	1.08	6.32	6.29	10.02	1.26	−0.84
Skewness	0.22	−0.24	−0.58	2.55	2.51	2.76	0.29	−0.38
Range	86	0.37	13.81	3.5	44	72	573	90
Minimum	30	0.22	8.66	0.9	10	18	2	4
Maximum	116	0.59	22.47	4.4	54	90	575	94
Count	193	193	193	193	193	193	192	193
Confidence Level (95.0%)	1.71	0.01	0.34	0.10	1.10	1.44	14.06	3.00

**Table 2 medsci-13-00020-t002:** Comparison of stratified age groups.

Group	1	2	3	4	5	*p* Value *
Number (N) per group	27	31	54	30	50	
Age range (years)	≤30	31–45	46–60	61–75	≥76	0.06
INR	1.29	1.18	1.3	1.46	1.59	0.06
PT (seconds)	14.94	13.63	14.94	16.92	18.2	0.06
APTT (seconds)	29.41	28.16	31.44	35.43	29.3	0.04
Platelet count (×10^9^/L)	264.15	241.94	209.28	227.13	204.76	0.07
Sr. proteins (g/dL)	67.67	70.35	70.74	69.83	63.7	0.003
HCT (%)	39	41	39	36	37	0.01
eWBV (mPas)	15.81	16.53	16.4	15.88	14.89	0.008

* Multivariate analysis.

**Table 3 medsci-13-00020-t003:** Comparison of gender groups.

Variables	Gender Group	Mean	Std. Deviation	*p* Value
International normalization ratio	F	1.41	0.75	0.57
	M	1.35	0.59	
Prothrombin time	F	16.22	8.59	0.54
	M	15.52	6.77	
Activated partial thromboplastin time	F	30.94	10.96	0.73
	M	30.42	9.22	
Platelet count	F	240.29	91.56	0.02
	M	206.04	103.62	
Age	F	59.50	20.74	0.24
	M	55.96	21.28	
Serum protein level	F	66.38	10.84	0.02
	M	70.33	13.09	
Haematocrit level	F	0.38	0.06	0.07
	M	0.39	0.07	
Estimated whole blood viscosity	F	15.45	2.24	0.01
	M	16.32	2.48	

**Table 4 medsci-13-00020-t004:** Comparison of eWBV and haemostasis indices in dichotomous D-dimer groups.

Variables	D-Dimer	Mean	Std. Deviation	*p* Value
International normalization ratio	no	1.35	0.70	0.52
	yes	1.41	0.65	
Prothrombin time	no	15.62	8.27	0.59
	yes	16.23	7.08	
Activated partial thromboplastin time	no	29.25	7.45	0.03
	yes	32.55	12.60	
Platelet count	no	233.39	72.31	0.13
	yes	211.65	124.25	
Age	no	52.06	19.67	0.001
	yes	65.18	20.50	
Serum protein level	no	71.61	9.48	0.001
	yes	63.98	13.71	
Haematocrit level	no	0.40	0.06	0.001
	yes	0.36	0.06	
Estimated whole blood viscosity	no	16.67	1.79	0.001
	yes	14.83	2.66	

**Table 5 medsci-13-00020-t005:** Summary of Pearson correlation eWBV and haemostasis indices.

		INR	PT	APTT	PLT
International normalization ratio (INR)	Correlation				
	Sig. (2-tailed)				
Prothrombin time (PT seconds)	Correlation	0.992			
	Sig. (2-tailed)	<0.001			
Activated partial thromboplastin time (APTT seconds)	Correlation	0.485	0.487		
	Sig. (2-tailed)	<0.001	<0.001		
Platelet count (PLT 109.L)	Correlation	−0.127	−0.112	−0.114	
	Sig. (2-tailed)	0.08	0.123	0.116	
Estimated whole blood viscosity (eWBV mPas)	Correlation	−0.17	−0.15	−0.12	0.26
	Sig. (2-tailed)	0.017	0.032	0.094	<0.001

**Table 6 medsci-13-00020-t006:** Concordance among cases with repeat testing.

	Sex	Age	D-Dimer	eWBV	PLT	INR	PT	APTT
1a	M	58	Positive	11.28	88	1.1	12	26
1b	M	58	Positive	11.15	74	1.3	14	28
2a	M	66	Negative	22.47	107	1.3	14	40
2b	M	66	Negative	20.59	85	1.3	14	41
3a	F	71	Positive	13.07	273	1.4	16	66
3b	F	71	Positive	9.52	245	1.6	18	37
4a	M	15	Negative	15.71	258	1.1	13	25
4b	M	15	Negative	16.45	262	1.1	13	27
5a	F	86	Positive	15.39	210	1	11	25
5b	F	86	Negative	14.87	205	4	44	45
6a	M	79	Positive	9.19	53	1.2	13	27
6b	M	79	Positive	9.59	71	1.4	15	32
7a	M	13	Negative	14.05	208	1.2	14	28
7b	M	13	Negative	15.54	241	1.3	14	28
8a	F	30	Positive	9.81	301	1.7	19	38
8b	F	30	Positive	13.26	560	1.7	20	35

## Data Availability

All data generated or analyzed during this study are included in this published article.

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
