# Peer review of "Correlation Between Blood Coagulation Profile and Viscosity: Clinical Laboratory Observational Study"

_medsci, 2025, doi:10.3390/medsci13010020_

Round 1

Reviewer 1 Report

Comments and Suggestions for Authors

MS Needs the methodological solution for compensating the small sample size and the retrospective nature of the study.

The keywords from each cluster were arranged to form research topics, facilitated by ChatGPT-4.0. Initially, a specific context for bibliometric analysis was set in the interaction session using the command:  The data represent in this analysis may serve as a starting point for further evaluations, possibly on a wider and more uniformly distributed sample.  

To confirm the importance of the viscosity in the hemorheological risk,  the MS should include a thromboembolic risk score, which, based on data relating to the patient's characteristics and the pathology itself.

 Needs the validation process of the relationship between the estimated whole blood viscosity (eWBV) derived from haematocrit and serum protein levels.

Author Response

MS Needs the methodological solution for compensating the small sample size and the retrospective nature of the study.

  • A new section [2.7] has been added to improve on methodology and clarify – re: Limitations: Cognizance was given to sample size may be considered relatively small. Therefore, “practical strategies for getting the most out of data when sample size is small” were adopted (Hopkin et al., 2015)…

The keywords from each cluster were arranged to form research topics, facilitated by ChatGPT-4.0. Initially, a specific context for bibliometric analysis was set in the interaction session using the command:  

  • This is error of judgement. We have not used ChatGPT to conceptualize this work.
  • Indeed, as in the reference “…reported inverse relation of eWBV with INR (Nwose, Cann, et al., 2010)”, this is a study series that started more than a decade before the advent of ChatGPT.

The data represent in this analysis may serve as a starting point for further evaluations, possibly on a wider and more uniformly distributed sample.  

  • Thanks for the positive note

To confirm the importance of the viscosity in the hemorheological risk, the MS should include a thromboembolic risk score, which, based on data relating to the patient's characteristics and the pathology itself.

  • The last paragraph just befor conclusion has been expanded to read “Therefore, beyond contribution of empirical data to advance the medical science of blood stasis, this report advances eWBV as an alternative or complimentary pathology test applicable to a wide range of disease management. For instance, thromboembolic risk score assessment model is still questionable hence, limited in usage (Häfliger et al., 2024). Further studies would be necessary to compare eWBV as a haemorrheology predictor versus the risk score model in assessment of future thromboembolic events.”

 Needs the validation process of the relationship between the estimated whole blood viscosity (eWBV) derived from haematocrit and serum protein levels.

  • This was already in the 4th paragraph of the introduction.
  • The statement on lines 62 – 64 has been clarified accordingly – re: “Estimated whole blood viscosity (eWBV) is one test that has been advanced to extrapolate the value from routine haematocrit and serum protein (Tamariz et al., 2008). Indeed, the eWBV dates back over 30 years (Muldoon et al., 1995), and has undergone validation process including previous studies that have demonstrated the applicability of the medical science theory in diabetes (Nwose, Richards, et al., 2010), as well as proof of concept (E.U Nwose, 2010; Nwose & Butkowski, 2013; Nwose & Bwititi, 2022; Wi et al., 2023).”

 Other responses as per “Can be improved” checklist

  • Design: this is now expanded – re: “Design: This was a clinical laboratory observational study. Evaluation was designed to include descriptive approach, as well as comparisons and correlation. …also be described as cross-sectional observation or clinical audit, with a small cohort evaluation using archived clinical pathology data”
  • Methods: the revision includes expanding and extending from four to seven subsections. Inclusion criteria, method of eWBV measurement, and limitations are now added subsections.
  • Results: Table 2 is further improved with add-on row for number of participants per stratified age group. Further, various analyses are now highlighted at the beginning of their respective paragraphs
  • Conclusion: this is also further re-articulated – re: “This study identified haemostasis management as one area of clinical practice where laboratory methods would benefit from … the relevance lies in re-evaluation of eWBV for adoption and utilization as either alternative or complementary test to profile of APTT, INR and/or PT.”

Reviewer 2 Report

Comments and Suggestions for Authors

A study in the scientific literature shows that increased estimated whole blood viscosity (eWBV) predicts higher mortality in patients hospitalised with coronavirus 2019 (COVID-19) disease. Did the authors consider that those donating blood for the study may have been after illness COVID-19 disease? Could this have affected the blood viscosity test, blood clotting or other indices and if so, how?

Article ‘Correlation between blood coagulation profile and viscosity: An observational study in a clinical laboratory'. The manuscript presents results, including activated partial thromboplastin time (APTT) and prothrombin time (PT), using the international normalisation ratio (INR) and plasma D-dimer. In this publication, the authors undertook the task of investigating the correlation of the stasis index with haemostasis indicators. The study was conducted on 193 samples.

Point 1

A study in the scientific literature shows that increased estimated whole blood viscosity (eWBV) predicts higher mortality in patients hospitalised with coronavirus 2019 (COVID-19) disease. Did the authors consider that those donating blood for the study may have been after illness COVID-19 disease? Could this have affected the blood viscosity test, blood clotting or other indices and if so, how?

Point 2

Did the authors obtain the approval of the ethics committee for this study?

Point 3

Were any of the patients who donated blood on anticoagulants that might have had an effect on the INR results?

Point 4

Did any of the patients donating blood take any medication? if so, what effect did this have on the results obtained?

Point 5

How many patients per age group have been studied?

Point 6

Could the authors describe the blood viscosity measurement method. Make of viscometer, principle of operation, speed of rotation, temperature of measurement. How was the blood sample for the viscosity test taken?

Point 7

Whether the authors are of the opinion that the blood viscosity test can be used for the testing in patients with polycythemia.

Author Response

Point 1

A study in the scientific literature shows that increased estimated whole blood viscosity (eWBV) predicts higher mortality in patients hospitalised with coronavirus 2019 (COVID-19) disease. Did the authors consider that those donating blood for the study may have been after illness COVID-19 disease? Could this have affected the blood viscosity test, blood clotting or other indices and if so, how?

  • Not applicable. Data was pre-COVID

Point 2

Did the authors obtain the approval of the ethics committee for this study?

  • It is appropriately declared that “…study was conducted in accordance with the Declaration of Helsinki and granted waiver to use as de-identified data by the SWPS of NSW Pathology.”

Point 3

Were any of the patients who donated blood on anticoagulants that might have had an effect on the INR results?

  • Not applicable. Study did not identify any individual, nor were data collected from blood donation centres

Point 4

Did any of the patients donating blood take any medication? if so, what effect did this have on the results obtained?

  • Not applicable. Study did not identify any individual, nor were data collected from blood donation centres

Point 5

How many patients per age group have been studied?

  • Thanks for the observation. Table 2 is edited to include distribution of number across stratified age groups

Point 6

Could the authors describe the blood viscosity measurement method. Make of viscometer, principle of operation, speed of rotation, temperature of measurement. How was the blood sample for the viscosity test taken?

  • A new section [2.5] has been created to describe the blood viscosity measurement method – re: “Method of eWBV measurement: The blood viscosity in this study was determined by the extrapolation method, as previously published (E. U. Nwose, 2010). It is pertinent to note that this eWBV option had been compared with digital method used in reference laboratory (Nwose & Richards, 2011).”

Point 7

Whether the authors are of the opinion that the blood viscosity test can be used for the testing in patients with polycythemia.

  • This is now clarified on lines 199 – 200; re: “It has long being known that blood viscosity can be used to assess polycythemia complications (Tremblay et al., 2019), including but not limited to retinopathy (Foulds, 1987).”

Round 2

Reviewer 1 Report

Comments and Suggestions for Authors

N/A